# Inhibition of protein synthesis in M1 of monkeys disrupts performance of sequential movements guided by memory

Machiko Ohbayashi[1,2]*

[1]Department of Neurobiology, University of Pittsburgh School of Medicine, Pittsburgh, United States; [2]Systems Neuroscience Center, Center for the Neural Basis of Cognition, University of Pittsburgh, Pittsburgh, United States

**Abstract** The production of action sequences is a fundamental aspect of motor skills. To examine whether primary motor cortex (M1) is involved in maintenance of sequential movements, we trained two monkeys (*Cebus apella*) to perform two sequential reaching tasks. In one task, sequential movements were instructed by visual cues, whereas in the other task, movements were generated from memory after extended practice. After the monkey became proficient with performing the tasks, we injected an inhibitor of protein synthesis, anisomycin, into M1 to disrupt information storage in this area. Injection of anisomycin in M1 had a marked effect on the performance of sequential movements that were guided by memory. In contrast, the anisomycin injection did not have a significant effect on the performance of movements guided by vision. These results suggest that M1 of non-human primates is involved in the maintenance of skilled sequential movements.

## Introduction

The ability to perform a sequence of movements is a key component of motor skills, such as typing and playing a musical instrument. How the brain binds elementary movements together into meaningful actions has been a topic of much interest. The preparation for and generation of sequential movements is classically thought to depend on the supplementary motor area (SMA) and the pre-SMA (*Roland et al., 1980*; *Tanji and Shima, 1994*; *Gerloff et al., 1997*; *Shima and Tanji, 1998*; *Nakamura et al., 1998*; *Nakamura et al., 1999*; *Hikosaka et al., 2002*; *Picard and Strick, 2001*; *Dayan and Cohen, 2011*). According to this view, the primary motor cortex (M1) is thought to produce the patterns of muscle activity that are necessary to implement the motor plans generated by the premotor areas. There is growing evidence, however, that M1 of humans and non-human primates is involved in the acquisition and maintenance of sequential movements. For example, the activity of some M1 neurons appears to reflect the serial order of potential target stimuli (*Pellizzer et al., 1995*; *Carpenter et al., 1999*). Similarly, a large number of M1 neurons (~40%) and metabolic activity reflected aspects of learned movement sequences (*Lu and Ashe, 2005*; *Matsuzaka et al., 2007*; *Picard et al., 2013*). In addition, the functional connectivity of M1 neurons was suggested to be associated with sequential planning of movements (*Hatsopoulos et al., 2003*). These observations and others imply a more active role for M1 in the acquisition and maintenance of skilled movement sequences than previously thought (*Karni et al., 1995*; *Dayan and Cohen, 2011*). In fact, the results of imaging studies in humans underscore the importance of practice history and skill level for the activation of the motor areas (*Krings et al., 2000*; *Meister et al., 2005*).

However, causal experiments to test M1's involvement in the acquisition or maintenance of motor sequences have been challenging. M1 is critical for implementing motor output. Lesion or inactivation of M1 will abolish the motor commands to the spinal cord that generate muscle activity. Instead,

*For correspondence:
machiko@pitt.edu

Competing interests: The author declares that no competing interests exist.

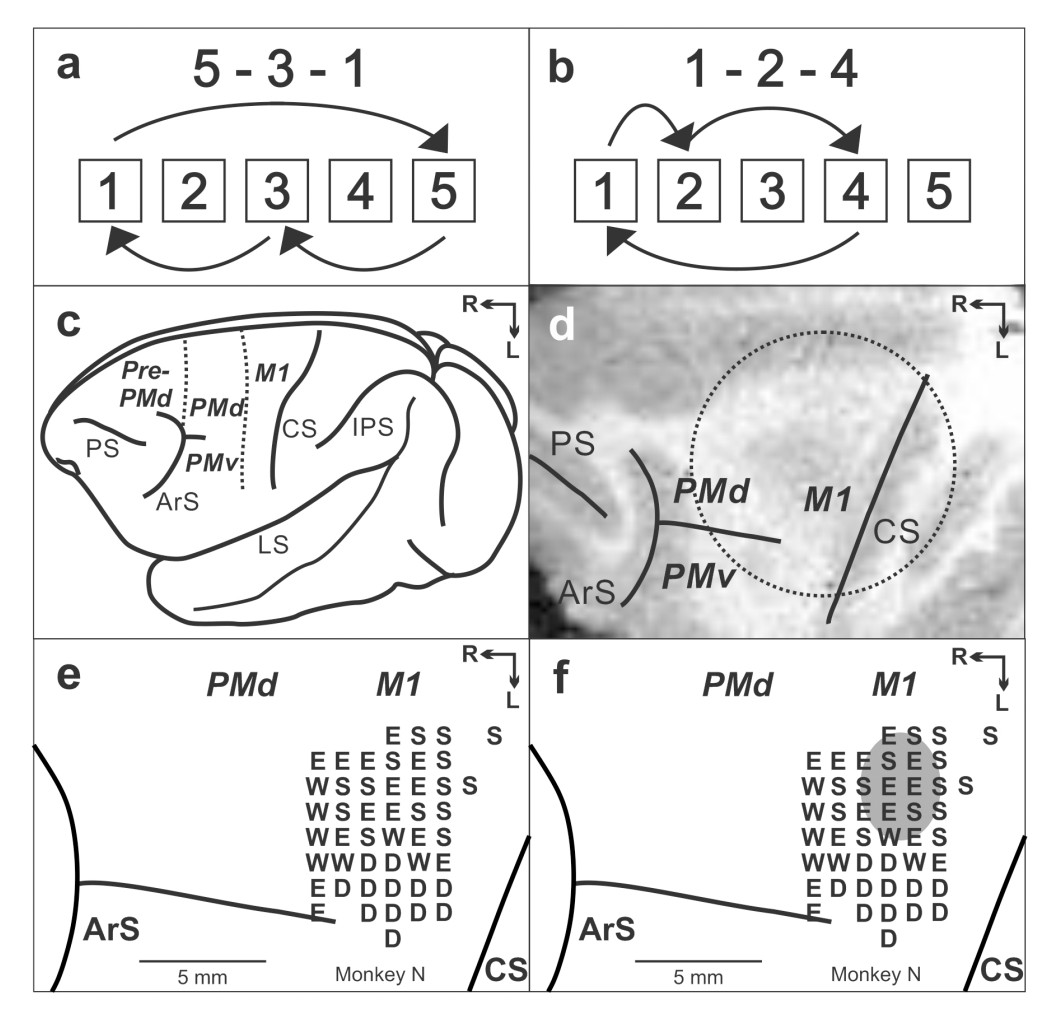

**Figure 1.** Task and cortical maps for monkey N. (**a, b**) Sequences in the Repeating task. (**c**) Lateral view of cebus brain. Dashed lines indicate the M1-PMd border and the pre-PMd-PMd border. PS: principal sulcus; ArS: arcuate sulcus; CS: central sulcus; IPS: intra parietal sulcus; LS: lateral sulcus; pre-PMd: pre-dorsal premotor cortex; PMd: dorsal premotor cortex; M1: primary motor cortex. R: rostral; L: lateral. (**d**) MRI image after the chamber implantation for monkey N. The dotted circle indicates the chamber outline. (**e**) Intracortical stimulation map from monkey N. Letter indicate the movements evoked at each site. S:Shoulder; E: Elbow; W: Wrist; D: Digit. f. Anisomycin solution was injected in the area indicated by a gray oval.

we manipulated protein synthesis in M1 to selectively disrupt information storage in this cortical area. This approach has been successful in the experimental dissection and analysis of other memory systems in rodents (*Davis and Squire, 1984*; *Nader et al., 2000a*; *Kleim et al., 2003*; *Luft et al., 2004*). In these studies, local injection of an inhibitor for protein synthesis, anisomycin, was shown to interfere with learning and maintenance of memory (*Davis and Squire, 1984*; *Nader et al., 2000a*; *Nader et al., 2000b*; *Kandel, 2001*; *Kleim et al., 2003*; *Dudai, 2004a*; *Luft et al., 2004*; *Dudai, 2012*; *Rudy, 2013*).

## Results

We trained two monkeys (*Cebus apella*) on two tasks. In one task, the monkeys performed sequential movements guided by memory (Repeating task) (*Figure 1a,b*). As a control task, the monkeys also were trained to perform reaching movements guided by visual cues (Random task, for details see Materials and methods). After the monkeys practiced each sequence for more than 100 training

days and became proficient with performing the two tasks, we made injections of the protein synthesis inhibitor, anisomycin, in M1 to test M1's involvement in maintenance of sequential movements after the extensive practice. The injections of aniomycin solution (100 mg/ml) were placed at sites in which intracortical stimulation evoked shoulder or elbow movements (*Figure 1c–f*, for details see Materials and methods). We analyzed the monkeys' behavior before and after the injection separately for each movement in each task. A movement from one target to the next is defined as a trial.

The anisomycin injections had a significant effect on the performance of the movements during the Repeating task (*Figures 2* and *3a,b*). The injections resulted in a significant increase in the number of errors (*Figures 2* and *3*; $\chi^2$ test, $p<0.05$) and a significant decrease in the number of predictive responses, an indication of sequence learning, (*Figure 3c*; $\chi^2$ test, $p<0.05$) during the Repeating task. In contrast, performance of visually guided movements during the Random task was not significantly disrupted ($\chi^2$ test, $p>0.05$).

The incorrect responses during the Repeating task can be categorized as two types: errors of accuracy and errors in direction. An accuracy error is a reach performed in the correct direction (e.g., to the right in move 1–5), but to an endpoint outside of the correct target (e.g., short of target 5). As shown in *Figure 2a*, before the injection, the monkey made correct contact to target 5 on 96% of the trials in the Repeating task (sequence 5-3-1). After the anisomycin injection, the number of incorrect responses increased dramatically. The monkey then made accuracy errors on 45% of the trials (*Figure 2a*).

A direction error was considered to be a reach performed in the direction opposite to the correct target (*Figure 2b*, $E_D$). For example, movement 2–4 requires a rightward movement from target 2 to target 4. As shown in *Figure 2b*, the monkey made a correct movement to target 4 on 93% of the trials in the Repeating task (sequence 1-2-4) before the injection. After the anisomycin injection, the animal moved his arm to the left from target 2 and made direction errors on 18% of the trials in the Repeating task (*Figure 2b*). This type of error suggests a deficit in selecting the movement component in the sequence. For the injection session illustrated in *Figures 2* and *3*, the error rate increased significantly for five of the six movements performed during the Repeating task.

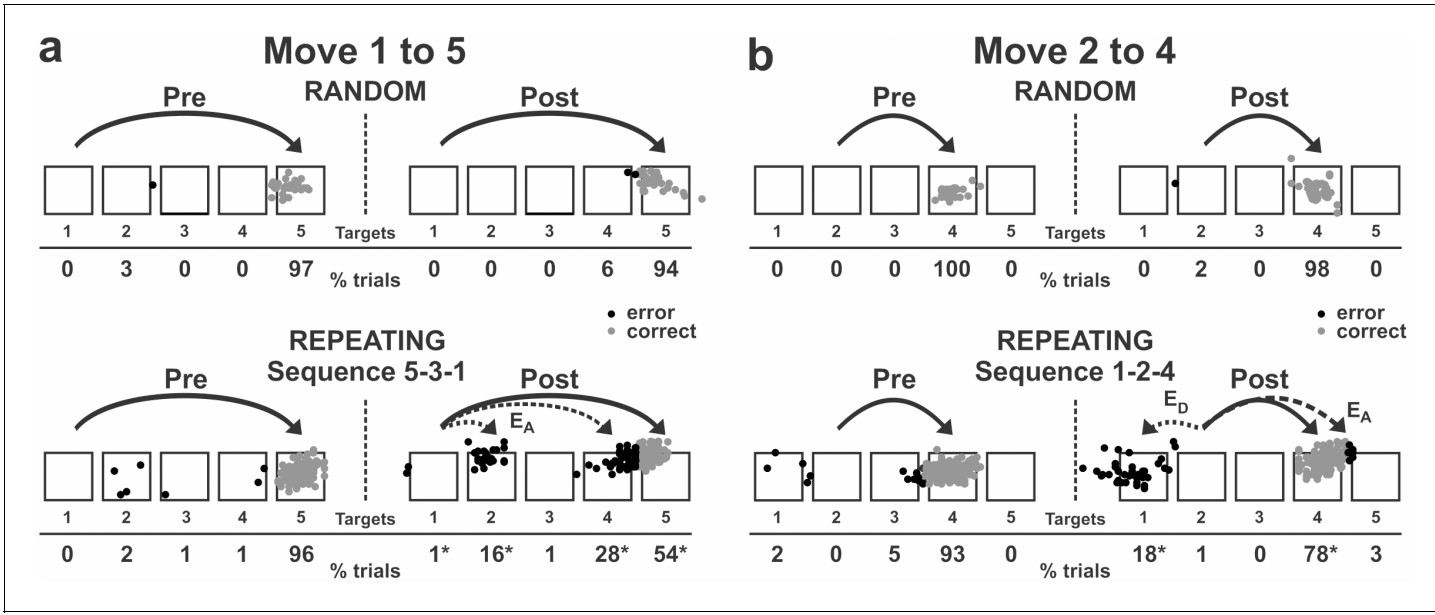

**Figure 2.** Reaching end points before and after anisomycin injection. Left: pre-injection; right: post-injection. $E_A$: Accuracy errors; $E_D$: Direction errors; gray dots: correct response; black dots: error response. The anisomycin injection was placed at sites shown in *Figure 1f* in monkey N. (a) Reaching end points for movement from target 1 to target 5. The monkey was performing sequence 5-3-1 during the Repeating task. (b) Reaching end points for movement from target 2 to target 4. The monkey was performing sequence 1-2-4 during the Repeating task. Percentage of trials ending in each target are given below the targets. Touches between targets were counted as touches to the closest target. *$p<0.05$.

The online version of this article includes the following source data for figure 2:

**Source data 1.** Contains numerical data plotted in *Figure 2a–b*.

Overall, we observed a significant increase in error rate during the Repeating task for all of six injection sessions ($\chi^2$ test, p<0.05; monkey N, n = 4; monkey R, n = 2). The effect of anisomycin was more pronounced for certain moves of the learned sequences (*Figures 3* and *4*). The affected moves varied between sessions. The error rate increased by an average of 42% for the most affected movements during the Repeating task (*Figure 4b*; paired t-test; p=0.002). The effect of anisomycin injections were consistent between monkeys (*Figure 4—figure supplement 1*). Given that performance of the Random task was unaffected, we attribute the performance changes observed in the Repeating task to the effect of anisomycin injection on memory. Overall, the effect of injections on the error rates was strong and consistent for the Repeating task and was consistently nonexistent for the Random task.

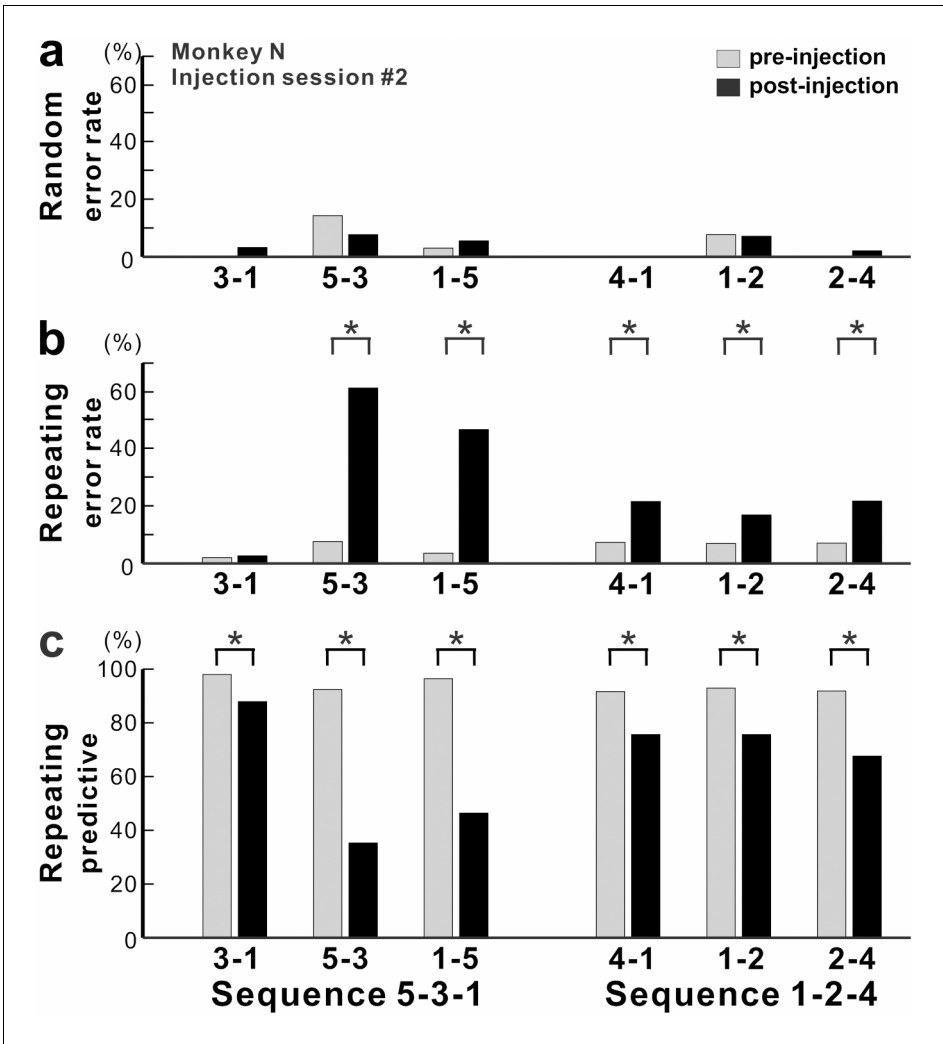

**Figure 3.** Effect of anisomycin injection on performance of the task. (**a-c**) The performance data for an injection session #2 in monkey N (*Figure 1f*, *Figure 2*). (**a**) Error rate in the Random task. Anisomycin injection did not have any effect on number of errors during the Random task ($\chi^2$ test, p=0.385 for move 3–1, df = 1; p=0.639 for move 5–3, df = 1; p=0.624 for move 1–5, df = 1; not significant for move 4–1, df = 1; p=0.957 for move 1–2, df = 1; p=0.371 for move 2–4, df = 1). (**b**) Error rate in the Repeating task. After the anisomycin injection, the number of errors increased dramatically in 5 of 6 movements in the Repeating task ($\chi^2$ test, p<0.001 for moves 5–3, 1–5, 4–1 and 2–4, df = 1; p=0.689 for move 3–1, df = 1; p=0.004 for move 1–2, df = 1). (**c**) Predictive responses. The percentage of predictive responses decreased significantly after the anisomycin injection ($\chi^2$ test, p<0.001 for all movements, df = 1).

The online version of this article includes the following source data for figure 3:

**Source data 1.** Contains numerical data plotted in *Figure 3a–c*.

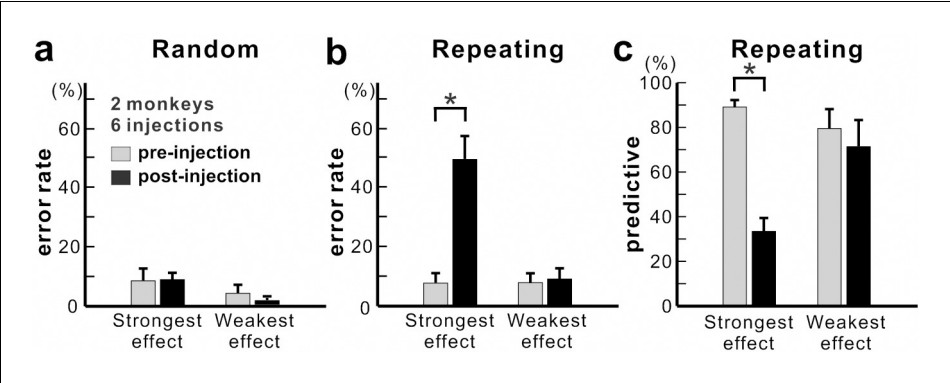

**Figure 4.** Population data for anisomycin injections. (**a**) Average error rate in the Random task. Anisomycin injections did not have an effect on the number of error responses in the Random task (paired t-test, six injection experiments, p=0.901 for strongest effect, df = 5; p=0.274 for weakest effect, df = 5). (**b**) Average error rate in the Repeating task. The error rate in the Repeating task increased significantly after the anisomycin injections (paired t-test, six injection experiments, p=0.002 for strongest effect, df = 5; p=0.497 for weakest effect, df = 5). (**c**) Average predictive responses in the Repeating task. The percentage of predictive responses decreased after anisomycin injections in M1 (paired t-test, p<0.001 for strongest effect, df = 5; p=0.419 for weakest effect, df = 5). *p<0.05.

The online version of this article includes the following source data and figure supplement(s) for figure 4:

**Source data 1.** Contains numerical data plotted in *Figure 4a–c*.
**Figure supplement 1.** Effect of anisomycin injections for each monkey.
**Figure supplement 1—source data 1.** Contains numerical data plotted in *Figure 4—figure supplement 1a–f*.
**Figure supplement 2.** Performance of the Repeating task from 3 days before the injection to 6 days after the injection for monkey N (**a–d**), monkey R (**e, f**) and both monkeys (**g, h**).
**Figure supplement 2—source data 1.** Contains numerical data plotted in *Figure 4—figure supplement 2a–h*.

During the Repeating task, there were trials when the monkey stopped the arm movement during midflight, redirected the arm and made a correct response within 800 msec after the presentation of the visual cue. These correct responses with longer RT were categorized as non-predictive responses (RT >150 msec) (see Materials and methods). We used the ratio of predictive/non-predictive trials as one of the indicators to assess the effect of injections on animal's task performance (*Figures 3* and *4*). As shown in *Figure 3c*, the ratio of predictive trials decreased from 92.4% to 35.3% for the most affected movement, from target 5 to 3, during the Repeating task ($\chi^2$ test, p<0.001). Overall, we observed a significant decrease in the number of predictive movements during the Repeating task for all of six injection sessions ($\chi^2$ test, p<0.05). The ratio of predictive responses decreased by an average of 55.7% for the most affected movements during the Repeating task (*Figure 4c*; paired t-test, p=0.002). A decrease in the number of predictive responses suggests an increase in the time for movement selection. In the non-predictive trials, it is possible that the monkey used the visual cue information to make a correct response since the animal's performance in the Random task was not affected by the injections.

After the injections, during the Repeating task, MTs significantly increased in 73.3% of the movements and RTs significantly increased for all the movements (two monkeys, six injections, t-test, p<0.05; averaged increase of MT: 45.82 msec, averaged increase of RT: 140.23 msec). The increase of RT and MT may reflect the monkeys' hesitation, uncertainty about the next target or change of the strategy. On the other hand, during the Random task, the injections did not cause a significant change in movement times for most of the movements in 5 of 6 injections (t-test, p>0.05). In three sessions, a small, but significant increase in movement time occurred only for one movement (t-test, p<0.05). Overall, during the Random task, RTs increased in sixty percent of the movements (t-test, p<0.05, averaged increase of RT: 84.61 msec; averaged increase of MT: 38.73 msec). In summary, the effects on RT and MT were strong and consistent within the Repeating task, but were weak and variable within the Random task. Anisomycin injection in M1 impaired performance of the Repeating task for 1–2 days after the injection ($\chi^2$ test, p<0.05). The task performance returned to the baseline after 1–2 days of training (*Figure 4—figure supplement 2*).

As control experiments, we inactivated M1 by injecting muscimol, a GABA agonist, into the shoulder representation of M1 in two sessions. We examined the performance of the animal on the Random and Repeating tasks as we did for anisomycin injections into M1. We found that M1 inactivations impaired performance on both the Random and Repeating tasks ($\chi^2$ test, p<0.05). The results indicate that M1 inactivation caused an indiscriminate deficit of motor production. Injection of saline into M1 did not have any significant effects on performance of both the Random and Repeating tasks ($\chi^2$ test, p<0.05).

## Discussion

In the present study, localized inhibition of protein synthesis in M1 resulted in a selective deficit in the performance of internally generated sequential movements during the Repeating task. This observation emphasizes the importance of M1 for the generation of sequential movements that are memory guided. Our results suggest that, although M1 is critical for movement production, it also is involved in the maintenance of skilled sequential movements.

Growing evidence showed that M1 is reorganized after extensive practice of sequential movements, yet our understanding of its neural basis after extensive practice is still limited. Human imaging studies reported that the volume of M1 is larger in professional musicians compared to amateurs or non-musicians suggesting that there is an effect of extensive practice on M1 structure (*Amunts et al., 1997*; *Gaser and Schlaug, 2003*; *Zatorre et al., 2012*; *Draganski and May, 2008*; *Herholz and Zatorre, 2012*; *Sampaio-Baptista and Johansen-Berg, 2017*; *Wenger et al., 2017*). Non-human primate studies also showed the effect of extensive practice of sequential movements on the neural and metabolic activity in M1 (*Matsuzaka et al., 2007*; *Picard et al., 2013*). After years of training, 40% of M1 neurons were differentially active during the performance of visually guided and memory guided sequential reaching (*Matsuzaka et al., 2007*). Uptake of 2DG in arm M1 was shown to be low in monkeys that performed highly practiced, internally generated sequences of movements (*Picard et al., 2013*). These observations imply a more active role for M1 in the planning and generation of sequential movements than previously thought. Our current observations also suggest that M1 is involved in sequential movements after extensive practice.

In rodent studies, the role of motor cortex in learning and maintenance of a skilled forelimb reaching task was tested by injecting an inhibitor for protein synthesis (*Kleim et al., 2003*; *Luft et al., 2004*). In these studies, the rats were trained to reach and grasp for a food pellet placed outside the cage (*Kleim et al., 2003*; *Luft et al., 2004*). Kleim and his colleagues reported that an injection of anisomycin into the motor cortex after 14 days of the task training disrupted the performance of the skilled forelimb task (*Kleim et al., 2003*). The injection of anisomycin not only disrupted the performance of the motor skill task, but also caused a significant reduction in synapse number and synapse size in M1 in vivo (*Kleim et al., 2003*). Luft and his colleagues reported that injection of anisomycin into the motor cortex after the 1st and 2nd days of training on the task disrupted the learning of the task (*Luft et al., 2004*). These studies suggested that the anisomycin injection interfered with the learning or maintenance of the forelimb motor skills such as reaching and grasping. In these studies, the effects of protein synthesis inhibition on the task performance were compared with motor production in well-established behaviors such as eating and walking. In addition, the rodents were trained for only 1–14 days prior to the injections in the previous studies (*Kleim et al., 2003*; *Luft et al., 2004*). On the other hand, in our study, we trained monkeys to perform sequential movements guided by memory (i.e. a complex motor skill) more than 100 days prior to the injection. In addition, the effect of anisomycin injection on the performance of the memory guided sequence was compared with the effect on a visually guided reaching task. Our results suggest that M1 is involved in maintenance of complex motor skills such as sequential movements after extensive practice.

The role of protein synthesis in learning and memory has been extensively studied especially in the context of fear conditioning of rodents (*Davis and Squire, 1984*; *Nader et al., 2000a*; *Nader et al., 2000b*; *Kandel, 2001*; *Dudai, 2004a*; *Dudai and Eisenberg, 2004b*; *Rudy, 2013*). De novo protein synthesis, during or shortly after initial training, is shown to be an essential step in consolidation of long-term memory (*Davis and Squire, 1984*). Moreover, the studies using anisomycin suggested the neural trace of memory may become labile upon retrieval, after which it may be reconsolidated (*Nader et al., 2000a*; reviewed in *Dudai, 2012* and *Rudy, 2013*). These studies using

anisomycin suggested that the neural trace may be destabilized through protein degradation and rebounded through protein synthesis during the reconsolidation (*Nader et al., 2000a*; *Nader, 2003*; *Sara, 2000*; *Lee et al., 2008*; *Rudy, 2008*; *Rudy, 2013*). The destabilized trace is proposed to be bi-directionally modified (i.e. weakened or strengthened), so that the memory can be 'updated' (*Rudy, 2008*; *Sara, 2000*; *Dudai, 2012*). Thus, when the protein synthesis inhibitor, anisomycin, was given during retrieval, the retrieved memory trace was lost as anisomycin prevents synthesis of the proteins needed to reconsolidate the memory trace (*Nader et al., 2000a*; *Lee et al., 2008*; *Dudai, 2012*). On the other hand, it is unclear how much these results can be generalized to other forms of memory. Our results showed that performance of well-trained sequential movements was impaired by injections of the protein synthesis inhibitor into M1 of monkeys during the repetitive training. These observations suggest that the neural traces for sequential movements may be repetitively strengthened over multiple sessions of practice through lingering protein synthesis in M1, which may lead to an increase of synaptic efficacy in M1 and slow improvement of the task performance. Further studies will expand our understanding of the mechanisms that support memory strengthening.

Furthermore, studies using brain slices showed that protein synthesis is required for long-lasting synaptic plasticity (late long-term potentiation) (reviewed in *Kelleher et al., 2004*) and spine-head enlargement and growth during learning (*Steward and Levy, 1982*; *Steward and Fass, 1983*; *Steward and Schuman, 2003*; *Tanaka et al., 2008*). Several observations indicate that the effects of protein synthesis inhibitors on long-lasting synaptic plasticity are likely to be a specific consequence of their translational blockade (*Linden, 1996*; *Huber et al., 2000*; *Beaumont et al., 2001*; *Kelleher et al., 2004*). These studies in rodents support our finding that inhibition of protein synthesis in M1 of monkeys interfered with performance of sequential movements guided by memory.

M1 is densely interconnected with the dorsal premotor cortex proper (PMd) (*Dum and Strick, 2005*). We recently demonstrated that inactivation of the PMd resulted in a selective deficit in the performance of internally generated sequences (*Ohbayashi et al., 2016*). We proposed that the PMd contributes to the internal generation of sequential movements through maintaining motor-motor associations. Our current results are consistent with the PMd functioning as a major source of input to M1 to guide the performance of internally generated sequences. Previous studies reported that extended practice on a sequence of movements resulted in dramatic alterations in the functional activation and neural responses of M1 (*Matsuzaka et al., 2007*; *Picard et al., 2013*). In addition, human imaging studies found a functional and structural change in M1 of professional musicians (*Elbert et al., 1995*; *Amunts et al., 1997*; *Gaser and Schlaug, 2003*; *Schwenkreis et al., 2007*; *Draganski and May, 2008*; *Herholz and Zatorre, 2012*; *Zatorre et al., 2012*; *Sampaio-Baptista and Johansen-Berg, 2017*; *Wenger et al., 2017*). These studies, along with our current results, suggest that M1 is involved in the maintenance of sequential movements after extensive practice. M1's specific contribution to the acquisition and maintenance of sequential movements needs to be further explored in future experiments.

## Materials and methods

The care of the monkeys and the experimental protocols adhered to the National Institutes of Health *Guide for the Care and Use of Laboratory Animals* and the U.S. Public Health Service Policy on Humane Care and Use of Laboratory Animals. All procedures used followed institutional guidelines and were approved by the Institutional Animal Care and Use Committee.

### Behavioral tasks

Two monkeys (Cebus apella, two males weighing 3.2 kg and 3.7 kg) were trained to perform two tasks, Random and Repeating tasks (*Figure 1a,b*) (*Matsuzaka et al., 2007*; *Picard et al., 2013*; *Ohbayashi et al., 2016*). In these tasks, the monkeys were required to make reaching movements to targets on a touch sensitive monitor with their right arms. When the monkey sat in front of a monitor, a task started and the outlines of targets were displayed on the monitor. The outlines of five targets were displayed in a horizontal row and identified as numbers 1 to 5 from left to right (*Figure 1a,b*). When the first trial of the day started, one of the targets was filled with yellow color. To make a correct response, the monkey was required to contact the filled target within 800 ms of

its coloring. Immediately after the animal's contact of the monitor the yellow fill disappeared and a new trial started.

In the Random task, new targets were presented according to a pseudo-random order 100 ms after contact of the correct target. Therefore, the monkeys performed the reaching movements guided by visual cues without having the inter-trial intervals. In the Repeating task, new targets were presented according to a repeating sequence of three elements. Three sequences were used in this experiments: 5-3-1-5-3-1 . . . and 1-2-4-1-2-4 . . . or 2-3-4-2-3-4 . . . (*Figure 1a,b*, two sequences for monkey N, one sequence for monkey R). New targets were presented 400 ms after contact of the correct target. This 400 ms delay promoted the performance of predictive responses in which the animal anticipated the next target in a sequence.

A liquid reward was given after every four to five correct responses. The monkey received a sound feedback for each response (correct hit: 1 kHz tone; error hit: 50 Hz tone). In the case of errors or no response, the trial was repeated. Each task was performed continuously in blocks of 200–500 trials that alternated in a session for a total of up to 4000 trials or until the monkey stopped working. Once initiated, the monkeys typically performed the task, touching one target after another without interruption until satiety. The monkeys were introduced to the Repeating task after the monkeys became proficient in the performance of the Random task after about 50 days of practice. Both monkeys became proficient with performing the two tasks after more than 100 training sessions. The injection experiments were performed after a monkey had more than 100 training sessions on each sequence.

## Surgery

We implanted a head restraint device, along with an MR compatible chamber for micro-injection, on an animal's skull using small screws and dental acrylic. All surgical procedures were performed under general anesthesia (1–3% isoflurane) using aseptic techniques. The animal received antibiotics and analgesics after surgical procedures. The chamber's placement over M1 was verified using structural MR images taken prior to and after the surgery (*Figure 1d*). When task performance returned to the pre-surgical level, we performed a craniotomy to access the cortex in the chamber.

## Intracortical microstimulation

We used intracortical microstimulation to identify the arm representation in M1 and to physiologically define the border between M1 and the PMd (*Dum and Strick, 2005*; *Ohbayashi et al., 2016*). We used glass-coated Elgiloy microelectrodes (0.6–1.5 MΩ at 1 kHz) to deliver intracortical microstimuli. A constant-current stimulator was used to deliver cathodal pulses (1–40 µA intensity, 10–20 cathodal pulses, 333 Hz, 0.2 ms duration) at a depth of 1500 µm below the cortical surface (*Dum and Strick, 2005*; *Ohbayashi et al., 2016*). Stimulus intensity was measured with a current monitor (Ion Physics). The motor response evoked by stimulation was determined by visual observation and muscle palpation. The response threshold was defined as the lowest stimulus intensity necessary to evoke a response on ~80% of the trials.

## Injections of pharmacological agents

We injected anisomycin (an inhibitor for protein synthesis), muscimol solution or sterile saline at 1.5 mm below the cortical surface using a 30 gauge cannula connected to a 10 µl Hamilton syringe (0.2 µl every 30–60 s). We prepared solutions of anisomycin (100 µg/µl in ACSF, pH 7.2–7.4) and muscimol (5 mg/ml in saline) from commercially available powders (Sigma-Aldrich, MO). The cortical sites of anisomycin injection in monkey N are displayed in *Figure 1e* (injection session No. 2). For each injection session, we injected total of 5 or 10 µl anisomycin solution into M1 (5 µl, n = 5; 10 µl, n = 1). To cover a large portion of the arm representation in M1, the anisomycin solution was injected at two to four sites in the arm representation area of M1 (*Figures 1f*, 1-3 µl at each site). Injection sites were placed more than 2 mm away from the border between M1 and PMd identified by microstimulation. Previous studies using monkeys reported that infusion of 3 µl of muscimol into cortex inhibited the activity of neurons within a diameter of 2–3 mm (*Nakamura et al., 1999*). Therefore, the effect of 2–3 µl of a pharmacological agent was assumed to be limited to M1. The cannula was left in place for ~5 min to allow diffusion of the solution and prevent its reflux and then removed. The animals were trained as usual the day before the injection. On the injection day,

animals were not trained before or after the injection. The effect on performance of the tasks was tested 20–24 hr after the injection of an inhibitor of protein synthesis. We did experiments with anisomycin injections four times in monkey N and two times in monkey R. Injections were spaced at least 7 days apart. In separate experiments, we also injected 1–3 µl of muscimol to inactivate M1 function to compare the results with anisomycin injections. The effect of muscimol on task performance was tested 20 min after the injection to allow for some diffusion of the chemical in the brain tissue before resuming the task. Saline (5 µl) was injected as a control in separate experiments. During a post-injection test session, blocks of Random and Repeating trials were alternated at frequent intervals (150–500 trials) to sample the animal's performance evenly as the effect of the test substance emerged and intensified. Measurement of performance on days prior to an injection and/or on trial blocks preceding an injection was used as the baseline for comparison with post-injection performance.

## Analysis of performance

For every trial, we recorded various task parameters and measures of performance. Recorded performance measures were: correct response, wrong hit error, no hit error or corrective response (correct responses that followed an error). From the times of touch screen hits, we derived the Movement Time (MT) and Response Time (RT) associated with each response. We defined MT as the interval between the release of contact from one target to touch of the next target. We defined RT during the Random task as the time between the presentation of a new target and contact of that target. We defined RT during the Repeating task as the time between two targets touches minus the delay time, 400 msec. We subtracted 400 ms to account for the delay in the cue presentation. This could result in a negative RT if the monkey moved quickly to the next target in the sequence before the presentation of a cue. RTs less than 150 ms were considered to be predictive (*Ohbayashi et al., 2016*). RTs less than 150 ms were chosen as a conservative cut-off for predictive responses as it is too fast for a simple reaction time to the visual cue.

The effect of an injection was assessed by examining the following: changes in the percentage of correct responses, types of incorrect responses, changes in RT and MT, and contact points on the touch screen, and the percentage of predictive responses (*Ohbayashi et al., 2016*). We excluded the following trials from analysis: 1) corrective responses because in this case the target was predictable as the error trial was repeated; 2) no-hit error responses because these few no-hit responses could be caused by an animal's low motivation; 3) trials during the Random task with RT < 150 ms because the monkey may have attempted to perform short RT trials in the Repeating sequence. For each movement, we used $\chi^2$ tests with Holm–Bonferroni's correction to examine the significance of changes in error rate and predictive responses. We used t-tests with Holm–Bonferroni's correction to examine changes of MT and RT. The effect of anisomycin was more pronounced for certain moves of the learned sequences. For each injection session, a movement with the largest increase in error rate during the Repeating task was defined as the movement with the strongest effect. Similarly, a movement with the smallest or no increase of error rate was defined as the movement with the weakest effect. The data of these movements during the Repeating task and the data of corresponding movements during the Random task were used for population analysis, average and SE of error rates, and percentage of predictive responses (see *Figure 4*).

## Acknowledgements

This work was supported by National Institutes of Health Grants R21NS101499 to MO and the Brain Sciences Project of the CNSI & NINS BS291006 to MO. I thank Dr. Peter L Strick for support and discussions; Drs. Nathalie Picard, Floh Thiels and Toshihiro Hayashi for suggestions and discussions; Dr. Donna Hoffman for proof reading; Moya Carrier for an animal training and assistance.

## Additional information

### Funding

| Funder | Grant reference number | Author |
| --- | --- | --- |
| National Institutes of Health | R21NS101499 | Machiko Ohbayashi |
| National Institutes of Natural Sciences | BS291006 | Machiko Ohbayashi |

This work was supported by National Institutes of Health Grants R21NS101499 to M. O. and the Brain Sciences Project of the CNSI & NINS BS291006 to MO. The funders had no role in study design, data collection and interpretation, or the decision to submit the work for publication.

### Author contributions

Machiko Ohbayashi, Conceptualization, Data curation, Formal analysis, Funding acquisition, Validation, Investigation, Visualization, Methodology

### Author ORCIDs

Machiko Ohbayashi (iD) https://orcid.org/0000-0003-2563-9049

### Ethics

Animal experimentation: The care of the monkeys and the experimental protocols adhered to the National Institutes of Health Guide for the Care and Use of Laboratory Animals and the U.S. Public Health Service Policy on Humane Care and Use of Laboratory Animals. All procedures used followed institutional guidelines and were approved by the Institutional Animal Care and Use Committee (IACUC protocol #13092292 & 1688737) of the University of Pittsburgh. All surgical procedures were performed under general anesthesia (isoflurane) using aseptic techniques, and every effort was made to minimize suffering.

### Decision letter and Author response

Decision letter https://doi.org/10.7554/eLife.53038.sa1
Author response https://doi.org/10.7554/eLife.53038.sa2

## Additional files

### Supplementary files

• Transparent reporting form

### Data availability

All data generated or analysed during this study are included in the manuscript. Source data files have been provided for Figures 2, 3, 4 and supplement figures.

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
