## [Decision Letter]

**Acceptance summary:**

The author has addressed the reviewers' previous concerns by performing new data analyses and by revising the manuscript. These changes greatly improved the manuscript. Although some weakness may remain (e.g. the spatial extent of anisomycin effects), addressing this experimentally would not be easy, and the manuscript in the current form already contains sufficient information for readers' evaluation. Overall, this short report provides an important finding that the primary motor cortex (M1) plays an important role in the execution and maintenance of sequential movements. The use of anisomycin is elegant and testing this in monkeys is significant.

**Decision letter after peer review:**

Thank you for submitting your article "Inhibition of protein synthesis in M1 of monkeys disrupts performance of sequential movements guided by memory" for consideration by *eLife*. Your article has been reviewed by three peer reviewers, including Naoshige Uchida as the Reviewing Editor and Reviewer #1, and the evaluation has been overseen by Richard Ivry as the Senior Editor. The following individual involved in review of your submission has agreed to reveal their identity: Shinji Kakei (Reviewer #3).

The reviewers have discussed the reviews with one another and the Reviewing Editor has drafted this decision to help you prepare a revised submission.

Summary:

This study examined the role of primary motor cortex (M1) in the performance of sequential movements of a forelimb in monkeys (*Cebus apella*). Previous studies have implicated the supplementary motor area (SMA) and the pre-SMA in the preparation and the generation of sequential movements but the role of M1 remains elusive. This is primarily because lesions or inactivation of M1 typically results in severe deficits of forelimb movement, and it is difficult to examine whether M1 is involved in a more specific aspect of movement. The present study used a protein synthesis inhibitor to examine the role of M1 in memory-guided forelimb sequential movements.

The authors injected a protein synthesis inhibitor, anisomycin, into M1, and motor performance was examined 20-24 hours after injection. Two monkeys performed a memory-guided forelimb sequential movement task (Repeating task) as well as a visually-guided movement task (Random task). In the Repeating task, after anisomycin injections, the monkeys made erroneous forelimb movements by moving to the direction opposite from the target or by moving to the correct direction but missing the target. In the control condition, the monkeys made a correct movement even before the next target appears ("predictive movements") in >90% of the trials. However, the number of predictive movements greatly decreased after anisomycin injections. By contrast, in the Random task, the performance was not impaired. Finally, injections of muscimol (an agonist of GABA-A receptor) into M1 impaired the performance in both the Repeating and Random tasks.

Overall, the reviewers thought that this study addresses an important issue of motor control, i.e. the role of the primary motor cortex (M1) in the maintenance of learned motor sequences. Using a protein synthesis inhibitor, the author was able to show the critical role of M1 in memory-guided forelimb sequential movements without perturbing a visually-guided forelimb movements. The results are potentially very interesting and may change the classical view on the role of M1 in higher order motor control. Generally, the manuscript is written clearly. However, the reviewers thought that some critical information is lacking and there are some technical concerns. We, therefore, would like to see your response before publication of this work in *eLife*.

Essential revisions:

1) The author injected anisomycin and observed its behavioral effects 20-24 hours after the injection. It remains unclear whether the author can exclude the possibility that the anisomycin diffused to the premotor cortex. This issue is important especially because the injection was made in the M1 area adjacent to the premotor cortex. Is it possible to estimate the protein synthesis in and outside the injected zone? If not, can the author estimate the spatial extent of protein synthesis inhibition in an informed manner?

2) The motor performance was tested 20-24 hours after anisomycin injection. In memory reconsolidation studies, it was shown that anisomycin produces amnesia if it is administered shortly after a memory retrieval, suggesting that what happened before anisomycin matters. First, it is, therefore, important to report whether and when the animals performed a task before anisomycin injections. Second, it is important to add some descriptions about recovery from anisomycin injection in the Results section. For instance, how were the animals trained for the task after an injection? How was recovery of their performance in general? Did the animals need to be trained after injection? In other words, did the effect of anisomycin disappear spontaneously or not? These questions appear to be important to discuss nature of "maintenance of sequential movements".

3) The author mentioned that monkey N was trained for two sequences and monkey R was for one sequence. They had more than 100 sessions on each sequence ("5-3-1-5-3-1…and 1-2-4-1-2-4" or "2-3-4-2-3-4"). Is it true that the sequence(s) that each monkey learned was fixed and never changed during the experimental course? If so, that means that the monkeys were over-trained on these sequences and had never relearned different sequences? Please make this point clearer in this manuscript. If the sequences were fixed, the conclusion of this manuscript is that protein synthesis in M1 is necessary for execution of over-trained motor repertories, not learning or its acquiring process. This, then, raises the question of how the inhibition of protein synthesis affect the performance without involving learning. Please discuss potential mechanisms in Discussion.

4) The authors mentioned that reaction time (RT) and movement time (MT) were measured but there is no description of their data. The author should report more detailed results on movements such as RT and MT, and potentially, the velocity profiles of forelimb movements. Are these variables affected by anisomycin or muscimol injections? It is important to know to what extent the forelimb movements are normal after anisomycin injections. Was there any time constraint on RT and MT for judgement of the correct or error trials? Please clarify whether and how often the monkey sometimes stopped the hand movement during the midflight and redirected the hand to the correct target. How was the judgement made in such case? Such details of the experimental design are difficult to follow in the current manuscript.

---

## [Author Response]

Essential revisions:1) The author injected anisomycin and observed its behavioral effects 20-24 hours after the injection. It remains unclear whether the author can exclude the possibility that the anisomycin diffused to the premotor cortex. This issue is important especially because the injection was made in the M1 area adjacent to the premotor cortex. Is it possible to estimate the protein synthesis in and outside the injected zone? If not, can the author estimate the spatial extent of protein synthesis inhibition in an informed manner?

The effects reported here are assumed to be attributed to M1 for the following reasons. First, an injection site was placed more than 2 mm away from the border between M1 and PMd identified by microstimulation. Previous studies using monkeys reported that infusion of 3 µl of muscimol into cortex inhibited the activity of neurons within a diameter of 2-3 mm (Nakamura et al., 1999). Therefore, the effect of 2-3 µl of a pharmacological agent was assumed to be limited to M1.

Second, we have seen different effects of anisomycin injections between M1 and PMd. In two separate experimental sessions, we injected anisomycin (3 µl, 100 mg/ml) into the shoulder representation of PMd. We examined the animal's performance on the Random and Repeating tasks as we did for anisomycin injections into M1. In one case, the anisomycin injection in the PMd impaired performance of both the Random and Repeating tasks (for move 1-5,% correct decreased from 78.0% to 31.6% during the Random task and decreased from 96.0% to 11.9% during the Repeating task, χ^2^ test, *p* < 0.001). In the other case, anisomycin injection in PMd did not have a statistically significant effect on performance of the tasks. In contrast, we observed that anisomycin injections in M1 impaired performance of the Repeating task in all of six injection sessions in M1 (χ^2^ test, *p* < 0.05).

These findings suggest that the effects reported here are attributable to M1. The above information has been added to the subsection “Injections of pharmacological agents”.

2) The motor performance was tested 20-24 hours after anisomycin injection. In memory reconsolidation studies, it was shown that anisomycin produces amnesia if it is administered shortly after a memory retrieval, suggesting that what happened before anisomycin matters. First, it is, therefore, important to report whether and when the animals performed a task before anisomycin injections. Second, it is important to add some descriptions about recovery from anisomycin injection in the Results section. For instance, how were the animals trained for the task after an injection? How was recovery of their performance in general? Did the animals need to be trained after injection? In other words, did the effect of anisomycin disappear spontaneously or not? These questions appear to be important to discuss nature of "maintenance of sequential movements".

The animals were trained as usual the day before the injection. On the injection day, animals were not trained before or after the injection. The effect of anisomycin on task performance was tested the following day, 20-24 hours after the injection. This highlighted information has been added to the subsection “Injections of pharmacological agents”.

At the reviewer’s request, we analyzed performance of the Repeating task from 3 days before the injection to 6 days after the injection for monkey N (Figure 4—figure supplement 2A-D), monkey R (Figure 4—figure supplement 2E, F) and both monkeys (Figure 4—figure supplement 2G, H). Anisomycin injection in M1 impaired performance of the Repeating task for 1-2 days after the injection (χ^2^ test, *p* < 0.05). The task performance returned to the baseline after 1-2 days of training. Monkey N did not perform the task for 2 days between training day #3 and #4 (Figure 4—figure supplement 2A, B). The performance after the two days of non-training (training day # 4) was not significantly different from task performance before the injection (χ^2^ test, *p* > 0.05). We plan to perform additional experiments to investigate the influence of re-training on performance recovery. These results will be the object of a follow-up publication.

The above information has been added to the seventh paragraph of the Results.

3) The author mentioned that monkey N was trained for two sequences and monkey R was for one sequence. They had more than 100 sessions on each sequence ("5-3-1-5-3-1…and 1-2-4-1-2-4" or "2-3-4-2-3-4"). Is it true that the sequence(s) that each monkey learned was fixed and never changed during the experimental course? If so, that means that the monkeys were over-trained on these sequences and had never relearned different sequences? Please make this point clearer in this manuscript. If the sequences were fixed, the conclusion of this manuscript is that protein synthesis in M1 is necessary for execution of over-trained motor repertories, not learning or its acquiring process. This, then, raises the question of how the inhibition of protein synthesis affect the performance without involving learning. Please discuss potential mechanisms in Discussion.

Yes, it is true that the monkeys were over-trained on the same sequence(s). As stated in the Materials and methods, “The injection experiments were performed after a monkey had more than 100 training sessions on each sequence”. To be clearer, we added the following text in the Results section: “After the monkeys practiced each sequence for more than 100 training days and became proficient with performing the two tasks, we made injections of the protein synthesis inhibitor, anisomycin, into M1 to test M1's involvement in maintenance of sequential movements after the extensive practice.”

To address potential mechanisms of effect of protein synthesis inhibition on over-trained motor skill, we added the following paragraph to the Discussion section:

“The role of protein synthesis in learning and memory has been extensively studied especially in the context of fear conditioning of rodents (Davis and Squire, 1984; Nader et al., 2000a,b; Kandel, 2001; Dudai, 2004a,b; Kelleher et al., 2004; Rudy, 2013).performance[…] Further studies will expand our understanding of the mechanisms that support memory strengthening.”

4) The authors mentioned that reaction time (RT) and movement time (MT) were measured but there is no description of their data. The author should report more detailed results on movements such as RT and MT, and potentially, the velocity profiles of forelimb movements. Are these variables affected by anisomycin or muscimol injections? It is important to know to what extent the forelimb movements are normal after anisomycin injections. Was there any time constraint on RT and MT for judgement of the correct or error trials? Please clarify whether and how often the monkey sometimes stopped the hand movement during the midflight and redirected the hand to the correct target. How was the judgement made in such case? Such details of the experimental design are difficult to follow in the current manuscript.

To evaluate the effect of injections on motor production, we tested the error rate, RT and MT of the Random task (i.e. visually guided reaching) before and after the injection. We did not see significant increase of error rate during the performance of the Random task (χ^2^ test, *p* > 0.05) (Figures 2, 3 and 4). On the other hand, during the Repeating tasks, the error rate significantly increased for the affected movements after the injections (χ^2^ test, *p* < 0.05) (Figure 2, 3 and 4). Given that performance of the Random task was unaffected, we attribute the performance changes observed in the Repeating task to the effect of anisomycin injection on memory. Overall, the effect of injections on the error rates was strong and consistent for the Repeating task and was consistently non-existent for the Random task.

During the Repeating task, MTs significantly increased in 73.3% of the movements and RTs significantly increased for all the movements after the injections (2 monkeys, 6 injections, t-test, *p* < 0.05; averaged increase of MT: 45.82 msec, averaged increase of RT: 140.23 msec). For example, for the movement from target 1 to 5 of the Repeating ‘5-3-1' in Figures 2 and 3, MT increased from 145.72 msec to 205.68 msec and RT increased from -125.13 msec to 48.31 msec (t-test, *p* < 0.001). There were trials when the monkey stopped the hand movement during midflight, redirected the hand and made a correct response. The increase of RT and MT may reflect the monkeys’ hesitation, uncertainty about the next target or change of the strategy. On the other hand, during the Random task, the injections did not cause a significant change in movement times for most of the movements in 5 of 6 injections (t-test, p>0.05). For example, MT did not increase significantly for the movement from target 1 to 5 during the session No. 2 in Figures 2 and 3 (t-test, *p* = 0.341; 234.53 msec vs. 248.43 msec). In three sessions, a small, but significant increase in movement time occurred only for one movement (t-test, *p* < 0.05). Overall, during the Random task, RTs increased in sixty percent of the movements (t-test, *p* < 0.05, averaged increase of RT: 84.61 msec; averaged increase of MT: 38.73 msec). In summary, the effects on RT and MT were strong and consistent within the Repeating task, but were weak and variable within the Random task.

During the Repeating task, there were trials when the monkey stopped the hand movement during midflight, redirected the hand and made a correct response within 800 msec after the presentation of the visual cue. These correct responses with longer RT were categorized as non-predictive responses (RT>150 msec) (see Materials and methods). We used the ratio of predictive/non-predictive trials as one of the indicators to assess the effect of injections on animal’s task performance (Figures 3, 4). As shown in Figure 3C, the ratio of predictive trials decreased from 92.4% to 35.3% for the most affected movement, from target 5 to 3, during the Repeating task (χ^2^ test, *p* < 0.001). Overall, we observed a significant decrease in the number of predictive responses during the Repeating task for all of six injection sessions (χ^2^ test, *p* < 0.05). The ratio of predictive responses decreased by an average of 55.7% for the most affected movements during the Repeating task (Figure 4C; paired t-test; *p* = 0.002). In the non-predictive trials, it is possible that the monkey used the visual cue information to make a correct response since the animal’s performance in the Random task was not affected by the injections.

When the monkey did not make a response within 800 msec after the presentation of the visual cue, the trial was recorded as a ‘no hit’ error and the same trial was repeated as a corrective trial. The corrective responses were removed from the analysis because the target was predictable after an error. The non-hit error trials were removed from the analysis because they could be an error caused by low motivation. Typically, there were few or no no-hit trials in a session except at the very end of a daily session (subsection “Analysis of performance”).

This information has been added to the Results section.